# Potentially Health-Promoting Spaghetti-Type Pastas Based on Doubly Modified Corn Starch: Starch Oxidation *via* Wet Chemistry Followed by Organocatalytic Butyrylation Using Reactive Extrusion

**DOI:** 10.3390/polym15071704

**Published:** 2023-03-29

**Authors:** Oswaldo Hernandez-Hernandez, Lesbia Cristina Julio-Gonzalez, Elisa G. Doyagüez, Tomy J. Gutiérrez

**Affiliations:** 1Institute of Food Science Research (CIAL) (CSIC-UAM), Nicolás Cabrera 9, 28049 Madrid, Spain; 2Centro de Química Orgánica “Lora Tamayo” (CSIC), Juan de la Cierva 3, 28006 Madrid, Spain; 3Grupo de Materiales Compuestos Termoplásticos (CoMP), Instituto de Investigaciones en Ciencia y Tecnología de Materiales (INTEMA), Facultad de Ingeniería, Universidad Nacional de Mar del Plata (UNMdP) y Consejo Nacional de Investigaciones Científicas y Técnicas (CONICET), Av. Colón 10850, Mar del Plata B7608FLC, Argentina

**Keywords:** gluten-free food, modified starches, noodles, resistant starch, short-chain fatty acids, starch digestibility

## Abstract

Extruded spaghetti-type pasta systems were obtained separately either from native or oxidized starch prepared *via* wet chemistry with the aim of evaluating the effect of oxidation modification of starch. In addition to this, the butyrylation reaction (butyrate (Bu) esterification—short-chain fatty acid) using native or oxidized starch was analyzed under reactive extrusion (REx) conditions with and without the addition of a green food-grade organocatalyst (l(+)-tartaric acid) with the purpose of developing potentially health-promoting spaghetti-type pasta systems in terms of increasing its resistant starch (RS) values. These would be due to obtaining organocatalytic butyrylated starch or not, or the manufacture of a doubly modified starch (oxidized-butyrylated—starch oxidation followed by organocatalytic butyrylation) or not. To this end, six pasta systems were developed and characterized by solid-state ^13^C cross-polarization magic angle spinning nuclear magnetic resonance (CP MAS NMR) spectroscopy, degree of substitution (DS), attenuated total reflectance Fourier transform infrared (ATR/FTIR) spectroscopy, X-ray diffraction (XRD), thermogravimetric analysis (TGA), pancreatic digestion, free Bu content analysis and *in vitro* starch digestibility. The results obtained here suggest that starch oxidation hydrolytically degrades starch chains, making them more susceptible to enzymatic degradation by α-amylase. However, the oxidized starch-based pasta systems, once esterified by Bu mainly on the amylose molecules (doubly modified pasta systems) increased their RS values, and this was more pronounced with the addition of the organocatalyst (maximum RS value = ~8%). Interestingly, despite the checked chemical changes that took place on the molecular structure of starch upon butyrylation or oxidation reactions in corn starch-based spaghetti-type pasta systems, and their incidence on starch digestibility, the orthorhombic crystalline structure (A-type starch) of starch remained unchanged.

## 1. Introduction

Spaghetti-type pasta (noodle) is a widely consumed food in the world, which is manufactured industrially by extrusion from wheat semolina (starchy food ingredient containing gluten)/water mixtures [1]. However, all those foods based on wheat (including spaghetti-type pastas), barley, rye, spelt, triticale and some oat varieties are limited for people suffering from celiac disease [2,3], since gliadin and glutenin (generally known as constituents of wheat protein or gluten) induce this autoimmune disease, which affects around 1% of the world population [4,5,6]. Although its incidence is apparently “low”, at the public health level, its impact on the generation of other pathologies associated with celiac disease such as bone diseases (osteopenia and osteoporosis), colon cancer and malnutrition (anemia) is important [7]. For this reason, the nutritional intervention of celiac disease through food science and technology is key to the management of this disease [8]. In this context, cassava, chestnut, chickpea, corn, potato, rice, sorghum and unripe plantain flours have been recommended by diverse research groups as starchy matrices capable of potentially replacing gluten-containing starchy sources [9,10,11,12,13,14,15,16].

With all of the above in view, this study was directed to develop health-promoting spaghetti-type pastas for all type of consumers. In this sense, corn (*Zea mays*) starch was selected as gluten-free feedstock [15]. In addition, corn starch, being dominated by the type A crystalline structure, can modulate starch digestion, since the linear amylopectin branches within the A-type crystalline lamellae can hinder and resist the enzymatic action of α-amylase [17]. The latter is very valued for the preparation of low-glycemic-index starchy foods, since this type of food also allows dietary intervention for people affected by diseases associated with metabolic syndrome such as cardiocirculatory diseases, overweight and type 2 diabetes, because slow starch digestion reduces the glycemic response in blood [18,19]. Apart from this, corn starch was particularly selected for this study, because its production is high worldwide [20].

On the other hand, the chemical, physical or enzymatic modification of starches with the aim of increasing their resistance to enzymatic action of α-amylase (type 4 resistant starch—RS4) is not something new for the manufacture of low-glycemic-index starchy foods [21]. Nonetheless, the dual chemical modification of starches based on novel processing and modification techniques is still on the rise with the purpose of increasing RS values [22]. With this in mind, oxidation and organocatalytic butyrylation (organocatalytically butyrate (Bu) esterification) reactions were selected in this investigation to increase the RS values of the designed spaghetti-type pastas.

Although starch oxidation has been widely studied [23], there are still important gaps with regard to its impact on the reduction or increase of RS values compared to their analogous native starches, since contradictory results have been reported. For example, Chung et al. [24] reported that oxidized starches had higher RS values than their analogous native starches, because the carboxyl (O−C=O) and carbonyl (C=O) groups formed in the starch structure could create steric hindrance for the enzymatic action of α-amylase [25]. In contrast, An [26] observed an opposite trend, which was attributed to the fact that oxidation hydrolytically degraded amylopectin and amylose (specifically in the amorphous regions of starch – amylose), thus facilitating the action of α-amylase. Likewise, the chemical modification of starch *via* organocatalytic esterification reaction using short-chain fatty acid (SCFA) precursors such as sodium butyrate (C_4_H_7_NaO_2_—NaBu) has been little studied in order to (1) increase the RS content [27,28,29], (2) raise the beneficial effect they cause on colon health [30,31,32,33], (3) regulate cholesterol in the blood [34], (4) modulate oxidative stress of colonic mucosa [35] and (5) enhance the growth of the colonic probiotic microbiota [36,37,38,39,40,41]. Additionally, the use of NaBu as a precursor of Bu to be grafted (esterified) onto the starch structure with the aim of obtaining REx-processed butyrylated starches has not yet been studied. Importantly, butyrylated starches are still in their infancy and more studies are required to be approved as “generally recognized as safe” (GRAS). These points are some novel aspects of this research.

It is worth noting that REx is not new as a starch modification methodology. Nevertheless, according to Ye et al. [42], REx can be regarded as a novel technique for obtaining RS. Moreover, as indicated above, the modification of starches is also not something novel with the purpose of raising the RS content. Notwithstanding, the manufacture of foods rich in organocatalytically modified starch has been barely studied, even *via* wet chemistry. Based on the best search made by the authors, no references were found on the manufacture of foods rich in organocatalytically butyrylated starch, obtained *via* REx, with the aim of increasing the positive effect of SCFAs formed and set free in the large gut.

REx as a one-step methodology for the simultaneous processing and modification of starch is of great industrial interest, due to its multiple advantages in terms of high energy efficiency, low operation time, low reagent and water consumption, and little equipment and personnel required [43,44,45]. This also highlights the perspective of this research regarding the large-scale production of health-promoting spaghetti-type pastas. Using a similar focus, Ye et al. [42], obtained high RS rice starch *via* REx employing citric acid as esterifying agent. Thus, this research combines three approaches to manufacture health-promoting gluten-free spaghetti-type pastas on a large scale: (1) dual starch modification, (2) REx as a one-step processing and modification technique and (3) obtaining organocatalytically butyrylated starches.

In particular, tartaric acid (C_4_H_6_O_6_—TAc) was used in this research work as a food-grade green organocatalyst, because (1) the United States Food and Drug Administration [46] approves its use as a GRAS food additive and (2) it is an organocatalyst that has shown a high rate of catalytic conversion with regard to other green food-grade organocatalysts to esterify (graft) Bu (butyric acid (BuAc) precursor—SCFA) onto the starch structure [27,32].

With everything described above, this investigation work raised and theorized the potential raise of RS values in spaghetti-type pasta systems based on butyrylated corn starch or organocatalytically doubly modified (oxidized-butyrylated) starch, processed by REx. With this in view, the structural and digestibility properties of extruded spaghetti-type pastas based on native (also known as regular) and oxidized corn starches were evaluated, followed by the butyrylation reaction with or without the addition of the selected organocatalyst (l(+)-TAc) with the objective of investigating (1) the effect of the oxidation modification of starch, (2) the effect of dual modification (oxidation-butyrylation) of starch and (3) the effect of the addition of the selected organocatalyst (l(+)-TAc) for the dual modification of starches. These objectives were established with the purpose of developing healthy spaghetti-type pastas in terms of increasing their RS values.

## 2. Experimental Method

### 2.1. Materials

Native and oxidized corn starch, l(+)-TAc, NaBu and distilled water were utilized as the carbohydrate polymer matrixes, organocatalyst, BuAc precursor and binder, respectively, for the fabrication of the spaghetti-type pastas. The ATR/FTIR spectra of the different feedstocks used to make the spaghetti-type pastas can be seen in the Appendix A. Native corn starch (NSt) was purchased from the distributor Dos Hermanos, Mark Ying Yang (Mar del Plata, Argentina). NaBu was obtained from Acros Organics (purity = 98.0%, code: 263190250, lot: A0386669, CAS: 156-54-7, Netherlands). l(+)-TAc (molecular weight = 150.09 g/mol) was procured from Fisher Chemical (purity ≥ 99%, code: T/0200/53, lot: 1870187, CAS: 87-69-4, Loughborough, UK). l-α-lysophosphatidylcholine from egg yolk (purity > 99%, Type I, powder, code: L4129-25MG, lot: SLBX1322, CAS: 9008-30-4, The Netherlands) and amylose from potatoes (code: A0512-250MG, lot: SLBT6849, CAS: 9005-82-7, USA) were bought from Sigma-Aldrich^®^ to quantify the total amylose content of the starches employed to make the spaghetti-type pastas. The total amylose content values were quantified following the protocol detailed by Gutiérrez [7], which is based on the enthalpy (ΔH) of creation of the amylose/l-α-lysophosphatidylcholine complex, resulting in total amylose content values of 19 and 17% for native and oxidized corn starch, respectively.

### 2.2. Starch Oxidation

Oxidized starch (OSt) was obtained *via* wet chemistry using a 20% *v/v* hydrogen peroxide (H_2_O_2_) solution as oxidizing agent. Starch oxidation was carried out strictly employing the method of Gutiérrez and Alvarez [47]. In summary, inside a reactor previously containing 4.2 L of distilled water, 1 kg of starch (dry basis) was mechanically mixed at 200 rpm for 15 min, and then the pH was adjusted to 9 by slowly adding a sodium hydroxide (NaOH) solution (2 M). After this, 126 mL of H_2_O_2_ (20% *v*/*v*) was added slowly. The reaction mixture was stirred at 200 rpm for 2 h at room temperature (25 °C). Next, the reaction mixture was neutralized to pH = 7 using a 2.5% hydrochloric acid (HCl) solution. Subsequently, the starch suspension was centrifuged at 1500 rpm for 15 min, and the supernatant solution was removed. The starch was resuspended and washed by adding distilled water, followed by centrifugation. This procedure was performed three times. Finally, the wet starch was dried in an oven at 45 °C for 24 h. Once this was complete, the dry oxidized starch was milled and passed through a 60-mesh sieve. Thereafter, the percentage of carbonyl (C=O) and carboxyl (O−C=O) groups on the OSt was determined following the protocol informed by Yi et al. [48]. In brief, the percentage of carboxyl (O−C=O) groups was determined by constant mechanical mixing of 2 g of oxidized starch and 40 mL of 0.1 M HCl for 30 min. Subsequently, the suspended starch was precipitated by adding approx. 80 mL of methanol, and then centrifuged to decant the supernatant. The moist starch was transferred to a beaker, and 300 mL distilled water was then added. Next, the new starch suspension was gelatinized by heating in a boiling water bath with constant stirring for 15 min. The hot starch suspension was adjusted to a volume of 450 mL with distilled water, and the resulting suspension was titrated with a 0.1 M NaOH solution utilizing phenolphthalein as indicator. A blank was made using the native starch. The carboxyl content was expressed as the number of carboxyl groups *per* anhydroglucose unit (AGU):(1)$$Carboxyl\;groups\;(\%)=\frac{{Mw}_{ag}\times M_{NaOH}\times (V_{NaOH}-V_{B})}{\left(W\times 1000)-(36\times M_{NaOH}\times (V_{NaOH}-V_{B})\right)}$$

Mw_ag_ = molecular weight of the anhydroglucose units (162 g/mol)

M_NaOH_ = concentration of the standardized NaOH solution (0.1 M)

V_NaOH_ = volume of NaOH used for the sample (mean value, mL)

V_B_ = volume of NaOH utilized for the blank (mean value, mL)

W = sample weight (g)

To determine the content of carbonyl (C=O) groups, about 1 g of the starch sample and 40 mL of distilled water were mixed in a beaker. The starch suspension was then gelatinized in a boiling water bath utilizing stirring for 20 min. The sample was allowed to cool to 40 °C, the pH was adjusted to 3.2 using a 0.1 M HCl solution, and then 25 mL of hydroxylamine reagent was added. The beaker was stoppered and placed in a 40 °C water bath for 4 h with occasional shaking. The excess hydroxylamine was determined by rapidly titrating the reaction mixture with a standardized 0.1 M HCl solution. A blank determination was performed in the same manner with the hydroxylamine reagent alone. The hydroxylamine reagent was prepared by dissolving 5 g hydroxylamine hydrochloride in 20 mL of a 0.5 M NaOH solution before adjusting the final volume to 100 mL with distilled water. The carbonyl content was calculated as follows:(2)$$Carbonyl\;groups\;(\%)=\frac{\left(M_{HCl}\times (V_{B}-V_{HCl}))\times (({36\times carboxil\;groups\;(\%))+Mw}_{ag}\right)}{W\times 1000}$$

Mw_ag_ = molecular weight of the anhydroglucose units (162 g/mol)

M_HCl_ = concentration of the standardized HCl solution (0.1 M)

V_HCl_ = volume of HCl used for the sample (mean value, mL)

V_B_ = volume of HCL utilized for the blank (mean value, mL)

W = sample weight (g)

The percentages of carbonyl and carboxyl groups on the OSt were approx. 0.21 and 0.05%, respectively.

### 2.3. Formulation and Manufacture of the Spaghetti-Type Pastas

All the spaghetti-type pastas were done utilizing a 190:101 (g/g) ratio (starch:water), using either NSt or OSt. The effect of added SCFA on spaghetti-type pasta blends was assessed by adding 5 g of NaBu (2.63% *w*/*w* with regard to starch content) employing both starch matrices assessed (NSt or OSt) in the presence and absence of the organocatalyst (TAc). Meanwhile, the organocatalytic effect (TAc) was explored by adding 5 g of this reagent (2.63% *w*/*w* with regard to starch content) in the presence of NaBu. The percentage of NaBu and TAc utilized was based on earlier research carried out by other authors who analyzed the effects of reaction time on butyrylated corn starches yielded by an organocatalytic route [49]. Each spaghetti-type pasta system was manually premixed until homogeneous mixtures were manufactured. The mixtures were then processed by extrusion in a twin-screw extruder (double A, Mar del Plata, Argentina) with six heating zones. The temperature profile utilized was 90/100/105/110/120/120 °C with a screw rotation speed of 30 Hz and a feed rate of 15 g/min. These extrusion conditions were based on an earlier research by our research group [50]. The six resulting spaghetti-type pasta systems were identified as follows: NSt, NSt + Bu, NSt + Bu + TAc, OSt, OSt + Bu and OSt + Bu + TAc. Finally, the prepared spaghetti-type pastas were conditioned under a controlled relative humidity (RH, ~57%) atmosphere for a week at 25 °C before characterization.

### 2.4. Spaghetti-Type Pasta Characterizations

#### 2.4.1. Solid-State ^13^C CP MAS NMR Spectroscopy

A Bruker AV WB 400 spectrometer (^13^C 100.73 MHz), Bruker Nano GmbH, Berlin, Germany, was used to acquire the One-dimensional (1D) solid-state ^13^C CP MAS NMR spectra at 300 K utilizing a 4 mm triple channel probe head. The sample specimens were packed in a 4 mm diameter zirconia cylindrical rotor with Kel-F end-caps. The instrument was operated under the following conditions: radio frequency (RF) field strength = 90.9 kHz and standard ^1^H two-pulse phase modulation (TPPM) decoupling pulse length sequences (90°) = 2.75 μs. The record parameters were: contact time = 3 ms, acquisition time = 40 ms, recycle delay = 5 s, spin rate = 7 kHz and spectral width = 25 kHz. ^13^C spectra were initially referenced to an adamantane sample. Thereafter, ^13^C (*δ*_C_) chemical shifts were recalculated to Me_4_Si [for CH_2_ atom *δ*(adamantane) = 29.5 parts *per* million (ppm)].

#### 2.4.2. Determination of DS by Butyrylation

The DS of each spaghetti-type pasta system was quantified by titration following the methodology used by Li et al. [51]. Briefly, all dry spaghetti-type pasta systems (~150 mg dry basis—W = weight of sample utilized (mg)) were manually crushed and passed through a 60-mesh sieve, and then 5 mL of a 1 M NaOH solution were added to saponify all ester linkages. The samples were allowed to react for 24 h at ambient temperature, and then excess unreacted NaOH was quantified by titration with a standardized 0.5 M HCl solution (V_HCl_ = volume (mL) of HCl needed for sample titration). The percent esterified Bu (butyryl group—%Bu) was determined as follows [40]:(3)$$\%Bu=\frac{\left(V_{B}-V_{HCl}\right)\times M_{HCl}\times {Mw}_{Bu}}{W}100\%$$

The DS was then calculated by replacing Equation (1) into Equation (2):(4)$$DS=\frac{{Mw}_{ag}\times \%Bu}{{Mw}_{Bu}\times 100\%-(({Mw}_{Bu}-1\;{g\;mol}^{\;-1})\times \%Bu)}$$

M_HCl_ = concentration of the standardized HCl solution (0.5 M)

Mw_ag_ = molecular weight of the anhydroglucose units (162 g/mol)

Mw_Bu_ = molecular weight of substituent butyryl group (71.09 g/mol)

V_B_ = volume of HCl required for the blank (mean value, mL)

Average DS values ± standard deviations (SDs) were informed by analyzing in testing three samples for each pasta system.

#### 2.4.3. ATR/FTIR Spectroscopy

The FTIR spectra of the spaghetti-type pasta systems were recorded utilizing a Nicolet 6700 FTIR spectrometer (Thermo Scientific Instrument Co., Madison, Wisconsin, USA). FTIR spectra were obtained employing the single reflection horizontal ATR accessory Smart Orbit and a diamond crystal at an incident angle of 45°. The operating conditions were the following: 32 co-added scans at 4 cm^−1^ resolution in the spectral range between 4000 and 400 cm^−1^. An OPUS software v.7.0 was utilized to operate the instrument. Two replicates for spaghetti-type pasta system were scanned to verify good reproducibility. The short-range ordered structures (double helices) of starch were calculated and estimated with regard to amorphous starch structures as follows: the 1045 cm^−1^/1011 cm^−1^ (A_1045_/A_1011_) absorbance ratio from the deconvolved FTIR spectra [34], whist the hydrated starch structures were calculated as the 996 cm^−1^/1011 cm^−1^ (A_996_/A_1011_) absorbance ratio from the deconvolved FTIR spectra [34].

#### 2.4.4. Scanning Electron Microscopy (SEM)

All spaghetti-type systems were immersed in liquid nitrogen, and then mechanically cryo-fractured. After this, the sample pieces were mounted on bronze stubs to be sputter-coated with a thin layer of gold, utilizing an Ar^+^ ion beam. The sputtering process was realized at a sputter rate = 0.67 nm/min and an energy level = 3 kV for 35 s. This was performed to diminish the surface charge of the samples during the observation and to ensure electrical conduction. SEM images of the cryo-fractured surface of the spaghetti-type pasta systems were then obtained employing a JEOL JSM-6460 LV scanning electron microscope. The SEM instrument was operated under the following conditions: magnification = 1.0 k× and acceleration voltage = 15 kV.

#### 2.4.5. XRD

X-ray diffractograms were obtained following the protocol informed by Hernandez-Hernandez et al. [34]. In brief, the assessed spaghetti-type pasta systems were analyzed utilizing a PAN X’Pert PRO diffractometer (The Netherlands) equipped with a monochromatic Cu K_α_ radiation source (λ = 1.5406 Å). The instrument was operated at 40 kV and 40 mA, and the sample specimens were examined in the 2*θ* range between 5 and 33° at 1°/min. The percent crystallinity (X_c_) of spaghetti-type pasta systems were evaluated following the procedure described by Hernandez-Hernandez et al. [34].

#### 2.4.6. TGA

A thermogravimetric analyzer (Model TGA Q500, TA Instruments, Hüllhorst, Germany) was utilized to examine the thermal stability of the analyzed spaghetti-type pasta systems. The TGA curves were registered under the following conditions: constant heating rate = 10 °C/min, temperature range between 30 and 900 °C, and nitrogen flow = 30 mL/min. The masses of the sample systems ranged between 5.7 and 8.7 mg. Three replicates *per* spaghetti-type pasta system were assessed to assure repeatability. SDs were as low as 1% for all sample specimens tested, and characteristic curves for each system were informed.

#### 2.4.7. Pancreatic Digestion and Free Butyrate (Bu) Content Analysis

The stability of ester linkages created between starch and butyrate in butyrylated starch-based spaghetti-type pasta systems was studied employing pancreatin (lipase capable of cleaving the ester linkage of triacylglyceride-like substances) from porcine pancreas (8 × USP specifications) and bovine bile salts. This trial was then conducted strictly following the methodology employed by Hernandez-Hernandez et al. [34].

#### 2.4.8. *In Vitro* Starch Digestibility

The Megazyme Digestible and Resistant Starch Assay Kit (Megazyme, Wicklow, Ireland) was utilized to quantify the rapidly digestible starch (RDS—starch digestible in the first 20 min), slowly digestible starch (SDS—starch digestible between 20 and 120 min), total digestible starch (TDS—starch digestible within the first 4 h, the average residence time of food in the human small gut) and RS (indigestible starch after 4 h) contents. The samples were digested following the manufacturer’s instructions, which are based on the methodology suggested by Englyst et al. [52] using pancreatic α-amylase and amyloglucosidase.

### 2.5. Statistical Analysis

Statgraphics Plus 5.1 software (Manugistics Corp., Rockville, MD, USA) was utilized to perform the analysis of variance (ANOVA) tests to examine the data. Statistical analysis results were informed as mean values ± SDs. The changes between the mean values of the checked properties were contrasted using multiple-range Tukey’s test, and employing a significance level of 0.05.

## 3. Results and Discussion

### 3.1. Structural Analysis: ^13^C CP MAS NMR, DS, ATR/FTIR, SEM, XRD and TGA

Two solid-state ^13^C CP MAS NMR peaks, attributed to carbonyl groups, were evidenced at ~173 and ~183 ppm for all pasta systems evaluated (Figure 1). The first peak (~173 ppm) can be related to the newly formed ester linkage, whilst the second peak (~183 ppm) can be attributed to the unesterified Bu in the manner of BuAc [29]. This confirms that the butyrylation reaction occurred on the starch structure in all Bu-containing pasta systems in the presence and absence of the organocatalyst, and that unesterified Bu residues unavoidably persist in all pasta systems (Figure 1). Similar results were informed by Hernandez-Hernandez et al. [34] who manufactured noodles based on organocatalytically propionylated regular and waxy corn starch *via* REx. It should be particularly noted that the ^13^C NMR peaks at ~173 and ~183 ppm for the OSt-based pasta systems (OSt + Bu and OSt + Bu + TAc pasta systems) can also be attributed to the carbonyl groups of the OSt itself. This explains why these systems showed more intense peaks in the signals indicated above compared to the NSt-based pasta systems (NSt + Bu and NSt + Bu + TAc pasta systems). In addition, a ^13^C NMR peak located at ~15 ppm was also observed for all pasta systems tested. This peak can correspond both to the methyl groups of the unesterified Bu residues, as well as to the Bu esterified to starch. However, another ^13^C NMR peak between 31 and 33 ppm was also displayed for all pasta systems, which can only be related to the -CH_2_- linkages of the newly created ester groups. On the other hand, typical ^13^C NMR peaks located between 20 and 30 ppm can only be exhibited due to the butyrylation reaction given on the starch [29]. This confirms that the butyrylation reaction occurred on both NSt and OSt, thus obtaining spaghetti-type pasta systems based on butyrylated starch for the first case, and doubly modified (oxidized-butyrylated) for the second case.

With respect to the DS values determined by titration, the Bu-containing spaghetti-type pasta systems showed the following ascending order: OSt + Bu (0.67) < NSt + Bu + TAc (0.83) < NSt + Bu (1.34) < OSt + Bu + TAc (1.39). Similar DS values ranging from 1.45 to 1.54 were found by Tupa et al. [49] for butyrylated corn starch upon the butyrylation reaction *via* wet chemistry over 2.5 and 7 h. Thus, all the Bu-containing pasta systems were butyrylated, independently of the addition of the organocatalyst utilized (TAc). Considering the classification given by Ragavan et al. [27], all the pasta systems studied had intermediate DS values.

Higher DS values were exhibited for the NSt + Bu pasta system compared to the OSt + Bu pasta system. This fact can be explained; because the −OH groups available to give rise to the butyrylation reaction were oxidized to carbonyl groups (see Section 2.2). This reduced the available sites for the butyrylation reaction to occur, and as a consequence, lower DS values were noted for the OSt + Bu pasta system (DS = 0.67) compared to the NSt + Bu pasta system (DS = 1.34). These results also suggest a predominance of the butyrylation reaction on the amylose backbone (amorphous regions of starch) with regard to amylopectin (crystalline regions of starch). It is worth remembering that the amylose contents for NSt and OSt were 19 and 17%, respectively (see Section 2.1). Imre and Vilaplana [32] informed a similar trend for organocatalytically butyrylated corn starches obtained from ungelatinized corn starch synthesized by wet chemistry.

As expected, the effect of adding the organocatalyst increased the DS values in the OSt + Bu + TAc pasta system (DS = 1.39) compared to its analogous pasta system without the addition of the organocatalyst (OSt + Bu pasta system — DS = 0.67). However, an opposite effect was observed, when the NSt + Bu (DS = 1.34) and NSt + Bu + TAc (DS = 0.83) pasta systems were compared.

It should also be noted that the DS values determined by titration can be overvalued from 31 to 91% with regard to the DS values calculated by NMR [39], since reverse titration does not enable discrimination between carboxylic acid groups and correlated side groups [53]. Nonetheless, Nielsen et al. [39] found that for ungelatinized butyrylated potato and corn starches synthesized by wet chemistry, free of starting reagents, i.e. there are no other substances such as Bu unesterified (also named unreacted Bu, ungrafted Bu or free Bu) that may affect the determination of the DS values by NMR. Conversely, as confirmed by studying the solid-state ^13^C CP MAS NMR spectra of pasta systems based on butyrylated starch processed by REx, unesterified Bu residues were evidenced. This is inevitable in pasta systems processed using REx, thereby limiting the correct quantification of DS values by NMR in these systems.

ATR/FTIR spectra from all spaghetti-type pasta systems showed absorption bands typical of starchy foods (Figure 2A) [54,55]. Nonetheless, none of the spaghetti-type pasta systems studied here showed characteristic bands of carbonyl groups, even those OSt-based pasta systems (OSt, OSt + Bu and OSt + Bu + TAc pasta systems). Consequently, solid-state ^13^C CP MAS NMR spectra are better tools for elucidating ester linkage creation in esterified (butyrylated) starch-based pasta systems compared to ATR/FTIR spectra. Nevertheless, hydrated starch structures and short-range ordered structures can still be examined from the ATR/FTIR spectra in the studied pasta systems (Figure 2B, Table 1).

The number of short-range ordered structures for spaghetti-type pasta systems ranged between 0.58 and 0.73 (Table 1). In this context, several authors have indicated that these structures play a key role in increasing RS values [56,57,58,59]. However, how these structures influence starch digestibility is currently not well understood [60]. Hence, more studies should be conducted on this aspect, since in this study there was no clear trend that can be explained.

With regard to the hydrated starch structures, starch oxidation did not markedly (*p* ≤ 0.05) modify the number of hydrated starch structures in the OSt pasta system compared to the NSt pasta system (Table 1). Nor were the hydrated starch structures significantly altered (*p* ≤ 0.05) in the Bu-containing pasta systems in the absence of the organocatalyst (NSt + Bu and OSt + Bu pasta systems) compared to their respective analogous pasta systems (NSt and OSt pasta systems) (Table 1). However, the Bu-containing OSt-based pasta system in the presence of the organocatalyst (OSt + Bu + TAc pasta system — pasta system based on starch doubly modified (oxidized-butyrylated) *via* organocatalysis) showed the lowest value of the number of hydrated starch structures (Table 1). This is possibly attributed to the highest DS value (DS = 1.39) in this pasta system (OSt + Bu + TAc pasta system). Therefore, the Bu (SCFA with a hydrophobic character) esterified on the starch structure could reduce the hydrated starch structures. A reduction in the hydrophilicity of butyrylated wheat starches with regard to their respective native starch was also informed by Li et al. [40]. Nonetheless, the number of hydrated starch structures for the NSt + Bu + TAc pasta system was markedly increased (*p* ≤ 0.05) with regard to its analogous control pasta system (NSt pasta system) (Table 1).

As for the morphology of spaghetti-type pasta systems, all pasta systems can be regarded as mainly amorphous, since a complete starch gelatinization was observed without signs of retrograded starch particles (type 3 resistant starch—RS3) in any of the pasta systems analyzed (Figure 3). Hence, the RS content values obtained for the pasta systems developed in this study (see Section 3.2 Table 2) must be primarily related to starch modification (RS4). Beyond these observations, no morphological changes were evidenced in the evaluated pasta systems, as a result of the oxidation and butyrylation reactions, or the addition of the organocatalyst.

XRD patterns exhibited that the long-range ordered structure for all analyzed pasta systems was governed by the orthorhombic crystalline structure (A-type starch) [61]: allowed diffraction peaks were situated at 2*θ* = 12.6459° (6.9943 Å), 16.8958° (5.2433 Å), 19.5168° (4.5447 Å), 21.0853° (4.2100 Å), 22.3991° (3.9660 Å) and 23.4985° (3.7829 Å) (Figure 4) [40,49]. This type of structure is typical of corn starch-based foods (orthorhombic crystalline structure—A-type starch) [20]. On the other hand, all the pasta systems were principally amorphous, and had small crystalline fractions varying from 3.22 to 5.73% (Figure 4). The effect of starch oxidation also decreased X_c_ values in all OSt-based pasta systems (OSt, OSt + Bu and OSt + Bu + TAc pasta systems) compared to the NSt-based pasta systems (NSt, NSt + Bu and NSt + Bu + TAc pasta systems) (Figure 4). These results suggest that the oxidation reaction destroys the ordering of starch chains [62]. While, the Bu-containing pasta systems in the absence of the organocatalyst (NSt + Bu and OSt + Bu pasta systems) increased their X_c_ values with regard to their respective analogous pasta systems (NSt and OSt pasta systems) (Figure 4). Thus, the butyrylation reaction takes place principally in the amorphous regions of the starch (amylose), thereby raising the X_c_ values. This is consistent with the analysis realized from the DS values. A similar trend was also found by Lopez-Rubio et al. [63] for low-DS butyrylated corn starches obtained *via* wet chemistry. With regard to the effect of adding the organocatalyst on the crystalline structure, there are contradictory results, since the OSt + Bu + TAc pasta system had a higher X_c_ value (3.75%) than its analogous system without the addition of the organocatalyst (OSt + Bu pasta system − X_c_ value = 3.45%), whilst a lower X_c_ value (4.95%) was evidenced for the NSt + Bu + TAc pasta system compared to its analogous system without the addition of the organocatalyst (NSt + Bu pasta system − X_c_ value = 5.73%) (Figure 4). Therefore, the organocatalyst had a positive effect on the increase in the DS value in the OSt + Bu + TAc pasta system (DS = 1.39) compared to its respective analogous system without the addition of the organocatalyst (OSt + Bu pasta system − DS = 0.67), as a consequence of a greater esterification of Bu on the amorphous regions of starch (amylose), thereby raising the X_c_ values (Figure 4). In contrast, the organocatalyst appears to unexpectedly interfere with the butyrylation reaction in NSt-based pasta systems (NSt + Bu + TAc pasta system − DS = 0.83 *vs*. NSt + Bu pasta system − DS = 1.34) (Figure 4). It could be speculated that the organocatalyst used (TAc) establishes hydrogen (H)-bonding interactions preferentially on the amylose molecules [34], which are limited for the butyrylation reaction, and therefore the organocatalytic butyrylation reaction is conducted on the amylopectin molecules (crystalline regions of starch), thereby reducing the X_c_ values in the NSt + Bu + TAc pasta system compared to the NSt + Bu pasta system (Figure 4).

The TGA curves evidenced the following increasing order of thermal stability for the spaghetti-type pasta systems: NSt < OSt + Bu + TAc < NSt + Bu < OSt + Bu ≈ NSt + Bu + TAc < OSt (Figure 5). Starch oxidation slowed down the thermal degradation of the OSt pasta system compared to the NSt pasta system, i.e. the OSt pasta system was more stable than the NSt pasta system. A similar trend was found by Hebeish et al. [62] for perborate-oxidized starches. Possibly, oxidized starch increases H-bonding interactions between starch molecules, thus increasing thermal resistance. Meanwhile, the butyrylation reaction in the absence of the organocatalyst increased the thermal stability of the NSt-based pasta system (NSt + Bu pasta system) compared to its analogous control pasta system (NSt pasta system). This fact was even more accentuated with the addition of the organocatalyst (NSt + Bu + TAc pasta system *vs*. NSt pasta system). Hence, the addition of the used organocatalyst (TAc) could preferentially establish H-bonding interactions with the amylose molecules before giving rise to the organocatalytic butyrylation reaction, thus increasing thermal stability in the NSt + Bu + TAc pasta system compared to the NSt + Bu pasta system. This is in accordance with the discussion previously made from the correlative analysis between the DS values and the XRD patterns for referred pasta systems. Nonetheless, the butyrylation reaction in the absence of the organocatalyst showed an opposite trend for the OSt-based pasta system (NSt + Bu pasta system) compared to its analogous control pasta system (OSt pasta system). This was even more marked after adding the organocatalyst (OSt + Bu + TAc pasta system *vs.* OSt pasta system). In this context, a negative relationship between thermal stability and DS values was in general viewed for the pasta systems analyzed. A similar trend was informed by Li et al. [40] for butyrylated wheat starch. For such reason, a weakening of the starch structure could be occurring, as a result of a higher amount of esterified Bu groups on the starch structure. Considering that the esterified Bu groups on the structure have a hydrophobic character [32]. It can thus be suggested that higher DS values lead to lower thermal stability, as a result of the breaking of starch–starch H-bonding interactions.

### 3.2. Digestibility Analysis: Pancreatic Digestion, Free Butyrate (Bu) Content and In Vitro Starch Digestibility

No markedly significant changes (*p* ≥ 0.05) were found in the free Bu content values upon pancreatic digestion of the butyrylated starch-based pasta systems (NSt + Bu, NSt + Bu + TAc, OSt + Bu and OSt + Bu + TAc pasta systems) (Figure 6). This finding confirms that butyrylated starch-based pasta systems (organocatalyzed or not) can escape small-intestinal digestion. Curiously, DS values and free Bu content values showed an inverse relationship, independently of pancreatic digestion, thereby suggesting that Bu was esterified on the starch structure (Bu not available for quantification) (Figure 6). In particular, the OSt + Bu pasta system exhibited the lowest DS value (0.67) and the highest free Bu content value (0.84 mg/mL). In contrast, the NSt + Bu pasta system displayed the lowest free Bu content value (0.51 mg/mL) and almost the highest DS value (1.34). This behavior can be explained, since the -OH groups available to give rise to the butyrylation reaction were oxidized to carbonyl groups (see Section 2.2). This diminished the available sites for Bu esterification to occur, and as a consequence, the free Bu content increased in the OSt + Bu pasta system (0.84 mg/mL). This is in line with the discussion made from the DS values obtained in this investigation (see Section 3.1). With regard to the effect of adding the organocatalyst on the free Bu content, the NSt-based pasta system (NSt + Bu + TAc) reduced its DS values (0.83) and increased the free Bu content values (0.68 mg/mL) compared to the NSt-based pasta system in the absence of the organocatalyst (NSt + Bu pasta system — DS = 1.34 and free Bu content value = 0.51 mg/mL). However, an opposite trend was observed for OSt-based pasta systems (OSt + Bu pasta system *vs*. OSt + Bu + TAc pasta system). Therefore, adding the organocatalyst had a positive effect on promoting the butyrylation reaction for the OSt-based pasta system, but not for the NSt-based pasta system.

With regard to RS content, the OSt pasta system had lower RS content values compared to the NSt pasta system (Table 2). This fact is probably attributed to oxidation hydrolytically degrades amylopectin and amylose (particularly in the amorphous regions of starch—amylose), thereby facilitating the action of α-amylase [26]. This is line with the analysis made from the XRD patterns (see Section 3.1). While the butyrylation reaction, both with and without the addition of the organocatalyst, failed to significantly increase the RS content values in the NSt-based pasta systems (NSt + Bu pasta system − RS content = 3.9 ± 0.8% and NSt + Bu + TAc pasta system − RS content = 3 ± 2%) with regard to its analogous control pasta system (NSt pasta system − RS content = 3.3 ± 0.3%) (Table 2). This is consistent with the correlative discussion done previously from the DS values and the free Bu content values, i.e. the organocatalyst did not exert a positive effect on the promotion of the butyrylation reaction in the NSt-based pasta systems. In contrast, the butyrylation reaction in the OSt-based pasta system without the addition of the organocatalyst (OSt + Bu pasta system − RS content = 4 ± 1%) was able to markedly rise the RS content values in this pasta system made from doubly modified starch (oxidized-butyrylated starch-based pasta system) compared to the OSt pasta system (RS content = 1.6 ± 0.1%). This trend was even more marked when using the organocatalyst in the OSt + Bu + TAc pasta system (RS content = 8 ± 2%) compared to either the OSt + Bu pasta system (RS content = 4 ± 1%) or the OSt pasta system (RS content = 1.6 ± 0.1%). This also fits well with the correlative analysis performed above from DS values and free Bu content values, i.e. the organocatalyst showed a positive effect on the promotion of the butyrylation reaction in OSt-based pasta systems. Therefore, the dual modification (oxidation-butyrylation) of the starch and adding the organocatalyst in OSt-based pasta systems were able to satisfy the previously established objective of raising the RS values. Similar results were obtained by Khurshida et al. [64], who achieved a rise of RS values in cassava starch doubly modified by ultrasonication followed by acetylation.

Taking into consideration the classification given by Goñi et al. [65], the potential health benefits in terms of RS values for the spaghetti-type pasta systems studied here could be indicated as follows: low for the OSt pasta system, intermediate for the NSt, NSt + Bu, NSt + Bu + Tac and Ost + Bu pasta systems and high for the OSt + Bu + TAc pasta system. It should be noted that this order was assigned only considering the RS content values itself. However, chemical characteristics of this type of RSs could increase their potential health benefits, in terms of SCFAs released and absorbed in the colon and the potential benefit on the growth of the probiotic microbiota [36,66,67]. For this reason, more studies are required to clearly establish the health promotion of this type of pasta.

Regarding the variations in the RDS and SDS content values for the pasta systems studied here, a clear trend was not observed for these digestibility parameters, as to establish an adequate explanation for this. This is probably related to the poor understanding of the relationship between short-range ordered structures and starch digestibility.

## 4. Conclusions

The following points can be established as general conclusions from this research work: (1) all the butyrate (Bu)-containing corn starch-based spaghetti-type pasta systems were butyrylated, both in the absence and in the presence of the organocatalyst used (l(+)-tartaric acid—TAc). This was evidenced by solid-state ^13^C CP MAS NMR spectroscopy, (2) ^13^C NMR spectra elucidated that unesterified Bu and organocatalyst unavoidably persisted embedded in the REx-processed pasta systems, thereby limiting the proper determination of DS values. Notwithstanding, the estimated quantification of DS values obtained by titration, in general, agrees well with all the findings found in this research and (3) the orthorhombic crystalline structure (A-type starch) in corn starch-based spaghetti-type pasta systems remained unchanged regardless of the butyrylation or oxidation reaction under the conditions studied.

From the analysis of the results obtained, and based on the three main objectives previously established in the introduction, the following three specific conclusions of this work emerge:The butyrylation reaction apparently occurred on the amorphous regions of the starch (amylose), regardless of the starch used: native or oxidized starch, thus resulting in more crystalline pasta systems, in which their thermal resistance was reduced, due to the breaking of the starch–starch H-bonding interactions provoked by the esterification reaction of hydrophobic Bu groups on the starch structure.The oxidation reaction led to the production of more amorphous and thermal degradation-resistant spaghetti-type pasta systems compared to native starch (NSt)-based pasta systems. Regardless, the enzymatic resistance by α-amylase was lower in the oxidized starch (OSt)-based pasta system compared to the NSt-based pasta system, thus reducing RS values. Noteworthily, DS values by butyrylation were inferior in the OSt-based pasta system compared to the NSt-based pasta system in the absence of the organocatalyst. This suggests that the sites with the potential to be esterified (available -OH groups on the starch structure) were previously occupied (oxidized) by carbonyl (C=O) and carboxyl (O−C=O) groups.The addition of the organocatalyst (TAc) had a positive result on the increase of the DS values by butyrylation in the OSt-based pasta system, which caused increases in RS values in this pasta system made from doubly modified (oxidized-butyrylated) starch (starch oxidation *via* wet chemistry followed by organocatalytic butyrylation) and reduction of the free butyrate content values. Therefore, potentially health-promoting spaghetti-type pastas were prepared from organocatalytically doubly modified (oxidized-butyrylated) starch. Nevertheless, an opposite trend was observed once the organocatalyst was added to the NSt-based pasta systems. These findings suggest that organocatalyst–amylose hydrogen bonding interactions could limit the butyrylation reaction.

## Figures and Tables

**Figure 1 polymers-15-01704-f001:**
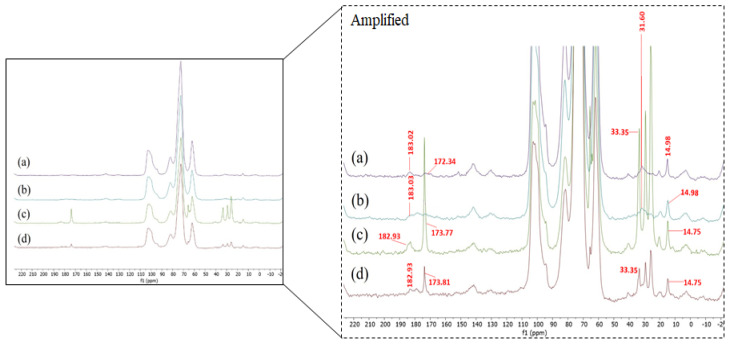
Solid-state ^13^C CP MAS NMR spectra from the different butyrylated starch-based spaghetti-type pasta systems studied: (**a**) NSt + Bu, (**b**) NSt + Bu + TAc, (**c**) OSt + Bu and (**d**) OSt + Bu + TAc.

**Figure 2 polymers-15-01704-f002:**
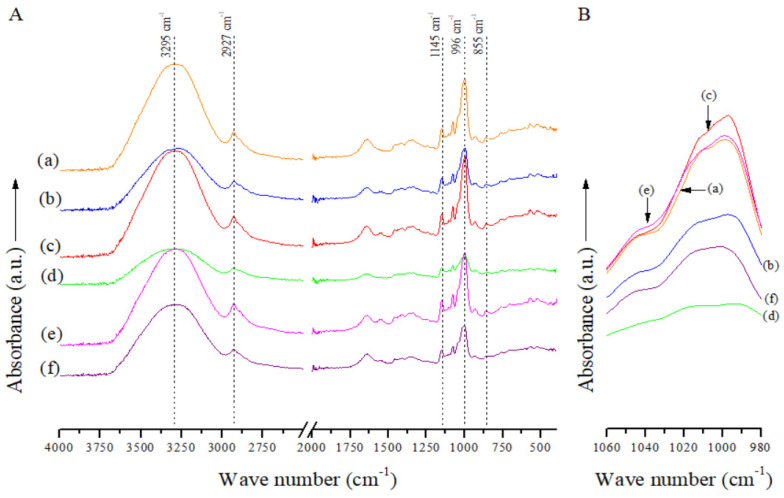
Panel (**A**) ATR/FTIR spectra from the different spaghetti-type pasta systems studied in the whole absorbance range. Panel (**B**) Region of normalized ATR/FTIR spectra for analysis of short-range ordered structures and hydrated starch structures. Spaghetti-type pasta systems: (**a**) NSt, (**b**) NSt + Bu, (**c**) NSt + Bu + TAc, (**d**) OSt, (**e**) OSt + Bu and (**f**) OSt + Bu + TAc.

**Figure 3 polymers-15-01704-f003:**
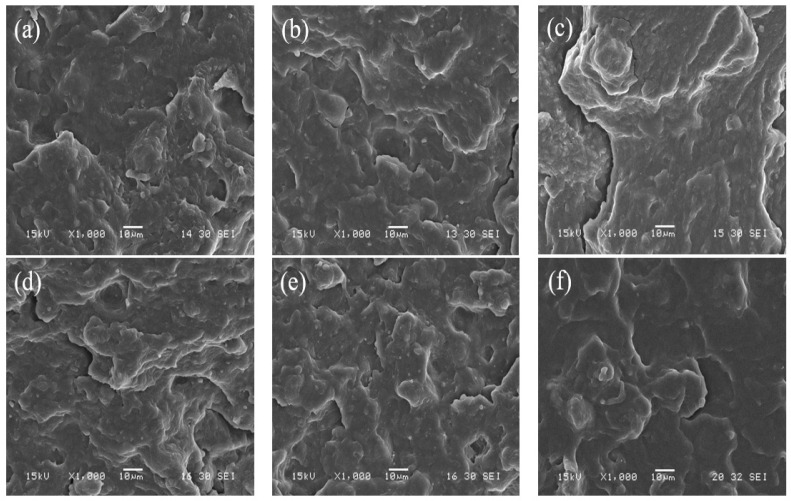
SEM micrographs of the cryogenic fracture surface from the different spaghetti-type pasta systems studied: (**a**) NSt, (**b**) NSt + Bu, (**c**) NSt + Bu + TAc, (**d**) OSt, (**e**) OSt + Bu and (**f**) OSt + Bu + TAc.

**Figure 4 polymers-15-01704-f004:**
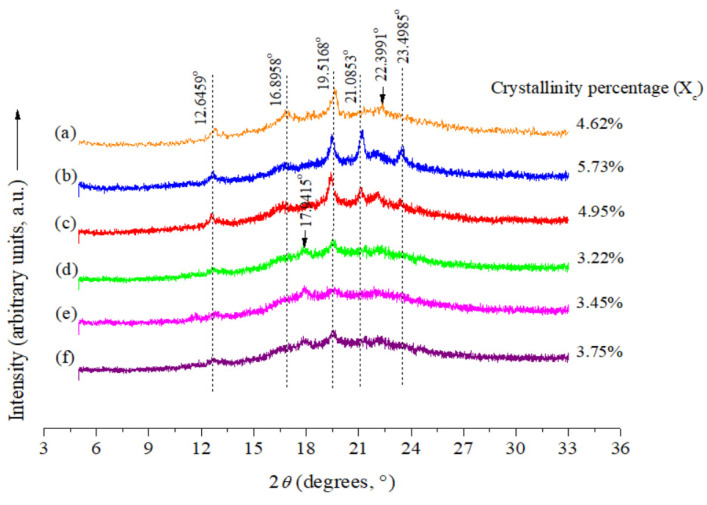
XRD patterns from the different spaghetti-type pasta systems studied: (**a**) NSt, (**b**) NSt + Bu, (**c**) NSt + Bu + TAc, (**d**) OSt, (**e**) OSt + Bu and (**f**) OSt + Bu + TAc.

**Figure 5 polymers-15-01704-f005:**
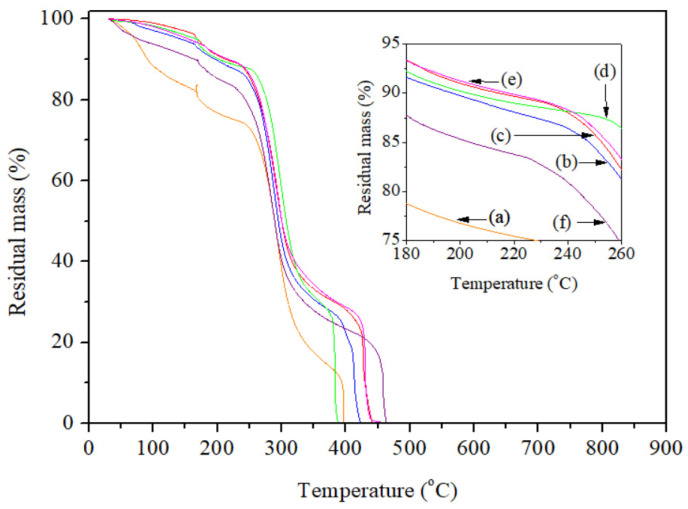
TGA curves from the different spaghetti-type pasta systems studied: (**a**) NSt, (**b**) NSt + Bu, (**c**) NSt + Bu + TAc, (**d**) OSt, (**e**) OSt + Bu and (**f**) OSt + Bu + TAc.

**Figure 6 polymers-15-01704-f006:**
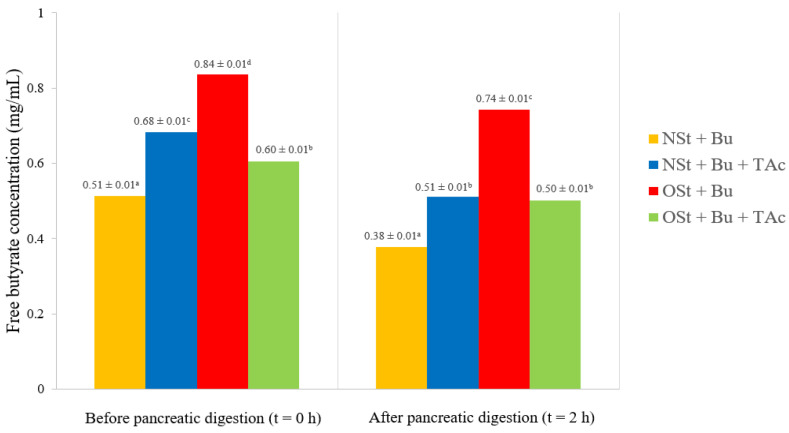
Free butyrate (Bu) content from the different butyrylated starch-based spaghetti-type pasta systems studied before and after pancreatic digestion. Different letters in the columns denotes statistically significant differences (*n* = 3, *p* ≤ 0.05).

**Table 1 polymers-15-01704-t001:** Structural analysis from the variations of the absorbances of the ATR/FTIR spectra of the different spaghetti-type pasta systems studied.

Pasta Systems	Short-Range Ordered Structures A_1045_/A_1011_	Hydrated Starch Structures A_996_/A_1011_
**NSt**	0.61 ± 0.01 ^b^	1.05 ± 0.01 ^a,b^
**NSt + Bu**	0.66 ± 0.01 ^c^	1.05 ± 0.01 ^a,b^
**NSt + Bu + TAc**	0.58 ± 0.01 ^a^	1.09 ± 0.01 ^c^
**OSt**	0.73 ± 0.01 ^d^	1.04 ± 0.01 ^a^
**OSt + Bu**	0.64 ± 0.01 ^c^	1.06 ± 0.01 ^a,b^
**OSt + Bu + TAc**	0.66 ± 0.01 ^c^	1.02 ± 0.01 ^a^

Different letters in the same column denotes statistically significant differences (*n* = 3, *p* ≤ 0.05).

**Table 2 polymers-15-01704-t002:** RDS, SDS, TDS and RS contents from the different cooked spaghetti-type pasta systems studied.

Samples	RDS (%)	SDS (%)	TDS (%)	RS (%)
**Cooked native corn starch ***	96 ± 3	11 ± 3	102 ± 2	0.8 ± 0.1
**NSt**	87 ± 5 ^a^	19 ± 1 ^e^	98.4 ± 0.2 ^b^	3.3 ± 0.3 ^a,b^
**NSt + Bu**	96 ± 1 ^c^	0.7 ± 0.1 ^a^	99 ± 4 ^b^	3.9 ± 0.8 ^a,b^
**NSt + Bu + TAc**	95 ± 2 ^c^	1.5 ± 0.6 ^b^	102 ± 3 ^b,c^	3 ± 2 ^a^
**OSt**	97 ± 2 ^c^	4.6 ± 0.2 ^c^	102.5 ± 0.7 ^b,c^	1.6 ± 0.1 ^a^
**OSt + Bu**	85 ± 4 ^a^	5.7 ± 0.2 ^d^	93 ± 1 ^a^	4 ± 1 ^a,b^
**OSt + Bu + TAc**	93.6 ± 0.9 ^b^	4 ± 1 ^c^	92.6 ± 0.6 ^a^	8 ± 2 ^c^

* Test control system. Different letters in the same column denotes statistically significant differences (*n* = 3, *p* ≤ 0.05).

## Data Availability

Transparency data associated with this article can be found in the online version at https://doi.org/10.17632/k259tfgbzj.2, accessed on 27 November 2022.

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
