# Peer review of "Potentially Health-Promoting Spaghetti-Type Pastas Based on Doubly Modified Corn Starch: Starch Oxidation via Wet Chemistry Followed by Organocatalytic Butyrylation Using Reactive Extrusion"

_polymers, 2023, doi:10.3390/polym15071704_

Round 1

Reviewer 1 Report

The authors deal with the problem of modification of corn starch by butyrylation reaction by reactive extrusion. This process allows for an increased value of RS starch.

The paper is very interesting, and the quality of the presentation is very high. Nevertheless, the authors should provide Figure 1 with higher quality. The proper NMR picks and sifts should also be provided in the paper or supplementary materials. The absorption bands typical for the starch compounds should be assigned to the proper groups.

After minor revision, the paper can be accepted. 

Reviewer 2 Report

The submitted manuscript assesses the effect of chemical modification of corn starch on resistant starch content in spaghetti type food product. The introduction highlights recent reports on the topic, but in my opinion does not justify the undertaken topic. The potential benefits of increasing RS content in gluten free product are indisputable, but the reason why such modification methods were used is not clear. The experimental material consisted of oxidized starch modified via hydrogen peroxide, whereas food grade oxidized starch according to FAO/WHO can be only treated with sodium hypochlorite (Annex 5 of the specification monograph prepared by JECFA). Moreover, the second modification performed (butyrylation) is also not approved to be used for food. The only food grade starch esters according to FAO/WHO are: phosphates, acetates, adipates, octenylsuccinates. In view of the above the aim to use such preparations for production of potentially health promoting food products seems a misguided idea. Also the obtained results of RS4 content are not overwhelming, as food grade preparations may reach around 50%, while in submitted paper it was below 10%. It is also not clearly indicated how the pasta was pre-treated prior each measurement, if and how was the cooking of the pasta performed, how long after preparation the samples were evaluated? The TGA analysis over 180 degrees is redundant, over that temperature carbohydrates starch to degrade and such temperatures are not used in process of pasta preparation. Other minor issues are hard to address at current point in view of the above and the lack of line numbering in themanuscript.

Reviewer 3 Report

Introdcution.

PLease include a section about teh physicochemical properties of starch

Materials and methods.

Starch oxidation was carried out strictly employing the method of Gutiérrez and Alvarez [47].

R:// You must include a brief explanation about the method.

Whereas, the percentage of carbonyl (C=O) and carboxyl (O−C=O) groups on the OSt was determined following the protocol informed by Yi et al. [48].

R:// The same, please include information.

The percentage of carbonyl and carboxyl groups on the OSt was approx. 0.21 and 0.05%, respectively.

R:// This  sentence is part of the results, please move it.

Results

The number of short-range ordered crystalline structures for spaghetti-type pasta systems ranged between 0.58 and 0.73

R:// From a crystallographic point of vie, the term “short-range ordered crystalline structures” does not have any sense. In starch there are two crystalline structures: orthorhombic and hexagonal (Rodriguez-Garcia et al. 2021.

XRD patterns exhibited that the long-range ordered structure for all analyzed pasta systems was governed by the type A crystalline structure:

R:// Dear Authors, starch has orthorhombic and hexagonal crystalline structures. If the starch has orthorhombic crystalline structure, it is classified as a A-type starch; if the starch has hexagonal crystalline structures, it is classified as B-type, and if the starch has both crystalline structures at the same time, it is called C-type. A, B, and C-type is the classification for starches, and the crystallines structures are orthorhombic and hexagonal Pinto et al 2020, Rodriguez-garica et al 2021)

Imbety et al 1998 pointed out that starch content monoclinic crystalline structures, but in term of crystallography both belong to the same spatial group but orthorhombic has smaller volumen, for this reason , the indexation is made using the orthorhombic cell. The powder diffraction file for monoclinic was not indexed by Imberty.

typical diffraction peaks situated at 2θ = 12.6o , 16.9o , 19.5o , 21.1o , 22.3o and 23.5o (Figure 4) [40,49].

R:// The X-ray diffraction does not have typical diffraction peaks, it has the allowed diffraction peaks  for each crystalline structure. The indexation for each peak for orthorhombic and hexagonal crystalline structures was reported by Rodriguez-Garcia et al 2021.

This type of structure is typical of corn starch-based foods

R:// What kind? Hexagonal or orthorhombic?

Although, a polymorphic transformation from the type A crystalline structure (monoclinic) to the type B crystalline structure (hexagonal) was observed, as a result of starch oxidation.

R: Although, a polymorphic transformation from orthorhombic crystalline structure to the hexagonal  crystalline structure was observed, as a result of starch oxidation.

This was evidenced by the rise in the intensity of the XRD peak located at 17.9o in all OSt-based pasta systems

R:// The change in the intensity of any X-ray diffraction peak, could be originated by several parameters, as the amount of simple on the holder, the X-ray technician expertise. But, in terms of X-ray diffraction the change in the intensity is associated with a change in the crystalline structure. If you have a crystalline transition, then you have new diffracted peaks that can be identified.

I believe that you must rewrite this section.

Pinto CC, Campelo PH, Michielon de Souza S: Rietveld-based quantitative phase analysis of B-type starch crystals subjected to ultrasound and hydrolysis processes. J Appl  Polym Sci 2020, 49529.

Rodriguez-Garcia, ME, Hernandez-Landaverde, MA, Delgado, JM., Ramirez-Gutierrez, CF, Ramirez-Cardona, M, Millan-Malo, BM, & Londoño-Restrepo, SM: Crystalline structures of the main components of starch. Current Opinion in Food Science 2021,37:107–111.

Imberty, A, Chanzy, H, Perez, S, Buleon, A, & Tran, V: New three-dimensional structure for A-type starchMacromolecules 1987, 20(10):2634-2636.

This paper showed for first time a partial indexation for the orthorhombic structure.

18. Takahashi, Y, Kumano, T, Nishikawa, S: Crystal structure of B-amyloseMacromolecules, 2004, 37(18): 6827-6832.

CONCLUSIONS: A polymorphic transformation from the type A crystalline structure (monoclinic) to the type B crystalline structure (hexagonal) was also evinced, as a result of oxidation modification.

R:// This is not true

This paper in the present form looks like a techinical report, mre analysis base don your results is necessary.

Typos

Typical diffraction peaks situated at 2θ = 12.6o , 16.9o , 19.5o , 21.1o , 22.3o and 23.5o (Figure 4) [40,49].

Change to: typical diffraction peaks situated at 2θ = 12.6, 16. , 19.5 , 21.o , 22.o and 23.5o (Figure 4) [40,49].

Reviewer 4 Report

The manuscript presented by the authors is quite interesting and novel for the development of gluten-free products such as pasta, which is one of the most popular foods worlwide. The study concerns about the preparation of spaguetti from modified corn starch which possesses a suitable content of resistant starch that promotes some health benefits. Authors achieved well the characterization of the pasta by using plenty analytical techniques. Therefore, the manuscript can be improved before publication after a minor revision would be done. The comments are in the following lines:

*Introduction section:
-More details are needed regarding health benefits of resistant starch or the consumption of organocatalytic butyrylated starch. Please use a cellular perspective.
-It is well known that reactive extrusion has environmental advantages. Please add information about this fact. The following reference is suggested:

*Results and discussion section:
-Please divide this section into subsections (3.1.1., 3.1.2., 3.2.1), showing each technique separately.
-According to SEM results: no morphological changes were found among pasta samples. It has been studied that microstructure is related to food texture. How can SEM results relate to the pasta's texture? Please explain.
-Table 2: please homogenize decimals.

*Conclusions are too long. It is better if the results are summarized. Authors need to remark the relevance of the study instead of recapitulate all the results.

Round 2

Reviewer 2 Report

The major concerns regarding experiment design were not addressed properly:

Response: Dear reviewer 2, We believe with all due respect that there must be some confusion, since according to United States Food and Drug Administration (USFDA, 2022) oxidized starch intended for food applications can be modified with hydrogen peroxide.

Indeed hydrogen peroxide can be used for process of starch oxidization, but at very low DS (carboxyl groups <0.1%). This type of starch is called bleached starch (INS 1403), whereas for oxidized starch (INS 1404) only sodium hypochlorite can be used. Details are in JECFA purity specification (which are international standard) https://www.fao.org/3/ca2330en/CA2330EN.pdf#page=74.

Response: As for the butyrylation modification, it is certainly not yet fully approved as a GRAS food additive. However, there are many more research groups indicating the benefits of this modification within the food industry. High-impact factor international journal-published research papers from other research groups, which are in this same line, are highlighted below:

A substance cannot be partially GRAS. I understand that butyrylation may be beneficial, and any further research is always and added value, but this type information MUST be stated and discussed in detail. Otherwise, the manuscript is misleading.

Response: On the other hand, the potential beneficial effect on health of starch based on its resistant starch content: negligible (<1%), low (1-2.5%), intermediate (2.5–5.0%), high (5.0–15% ) and very high (>15%) (Goñi et al., 1996). It suggests that the butylated starches studied here have intermediate potential to positively affect the health of diners. It should be noted that the classification given by Goñi et al. (1996) is widely accepted.

In my opinion, in view of the above (modification with hydrogen peroxide and butyrylation), the RS content should be extremely high to be considered and viable option to be considered. Since there are numerous other options available (even commercial clean label ones).

It is also important to note that the experimental part, section 2.3, indicates that the samples were extruded between 90 and 120 oC, i.e. they had a cooking process. Additionally, TGA analysis is typically performed even above its degradation temperature, since it is after 180 oC that differences associated with established chemical interactions can be observed. Finally, this manuscript was made using the template provided by the publisher.

For the above reason usually DSC is used for starch investigation rather than TGA.

Reviewer 3 Report

Dear Authors, again, starches that have orthorhombic crystalline structures are called A-type starches. If the starch has a hexagonal crystalline structure, it is called B-type. C-type starch is one that has both crystalline structures, as in the case of avocado starch (Esquivel-Fajardo et al 2022). XRD patterns exhibited that the long-range ordered structure for all analyzed pasta systems was governed by the type A crystalline structure

R:// A-type crystalline structure does not exist.

XRD patterns exhibited that the long-range ordered structure for all analyzed pasta systems was governed by the orhorhombic  crystalline structure, it means, that this starch can be clasified as A-type.

Situated at 2θ = 12.6o , 16.9o , 19.5o , 21.1o , 22.3o and 23.5º

R:// If you check the table proposed by Rodriguez-Garcia et al. (2021, you can find the position for each diffracted peak for orthorhombic crystalline structures. If your wave length is 1.5406 anstrons, the reported value for each peak has four significant digits.

Although, a polymorphic transformation from the type A crystalline structure (orthorhombicmonoclinic) to the type B crystalline structure (hexagonal) was observed

R: // This transformation is not possible; you may have both crystalline structures.

i.e. a combination of the type A and B crystalline structures, leading to the type C crystalline structure was observed.

R// This is not true; from a crystallographic point of view, it is not possible. If your sample has orthorhombic and hexagonal crystalline structures, then it can be classified as C-type.

This was evidenced by the rise in the intensity of the XRD peak located at 17.9o in all OSt-based pasta systems (OSt, OSt+Bu and OSt+Bu+TAc pasta systems)

R:// The intensity of any X-ray pattern depends on the amount of sample, structure, and form factors, among others.

The patterns in Fig. 4 are all orthorhombic; I cannot see the structural transformation, or the existence of both. These patterns must be improved.

A polymorphic transformation from the type A crystalline structure (orthorhombicmonoclinic) to the type B crystalline structure (hexagonal) was also evidencede

R: This is not true, and it is not posible.

I focused my attention now on the interpretation of the X-ray section. It needs to be revised by an expert. The paper needs a lot of work; you must change the abstract and conclusion.

Esquivel-Fajardo, E. A., Martinez-Ascencio, E. U., Oseguera-Toledo, M. E., Londoño-Restrepo, S. M., & Rodriguez-García, M. E. (2022). Influence of physicochemical changes of the avocado starch throughout its pasting profile: Combined extraction. Carbohydrate Polymers, 281, 119048.

Round 3

Reviewer 2 Report

No changes were made to the manuscript with regards to the second review report. The authors still mix the terms with regards to INS104 and INS1403 naming. Both preparations have different names, but both are prepared in process called oxidation. The authors have prepared INS1403 which is bleached starch and with that regards it can be used food, but the naming should be changed thought the manuscript. With regards to butyrylation, I partially understand the point of view of the authors, but would have to be thoroughly explained in the introduction. Currently out of 6 preparations investigated. With regards to TGA, the response from the second round leads back to the first report. Investigation of starch above 180°C form food technology perspective is pointless.

Reviewer 3 Report

In Fig. 2 some IR band appear in the figure, but no information about their origen was reported. I believe that IR analysis can be improved.

orthorhombic crystalline structure (A-type starch) [61]: allowed

diffraction peaks situated at 2θ = 12.6o, 16.9o, 19.5o, 21.1o, 22.3o and 23.5º

R:// Dear Authors, as you know, the wavelength for Cu is 1.5406 Anstrongs, it means that you must use at least four significant numbers.  12.6054 °

In Fig. 4 please change Intensity (a.u) by Intensity (Arbitr. Units) and 2θ (Degrees)

I believe that it is very important to recall in the abstract and conclusion, that there are not changes in the crystalline structure.

F you have the possibility to filter the X-rya patterns, it is possible that the peak identification will be better.

Typos:

orthorhombic crystalline structure (A-type starch) [61]: allowed

diffraction peaks situated at 2θ = 12.6o, 16.9o, 19.5o, 21.1o, 22.3o and 23.5º

R// Change to: orthorhombic crystalline structure (A-type starch) [61]: allowed

diffraction peaks situated at 2θ = 12.6xx, 16.9xxx, 19.5xxx, 21.1xx, 22.3xxx,  and 23.5º
